# Portal Vein Pulsatility: A Valuable Approach for Monitoring Venous Congestion and Prognostic Evaluation in Acute Decompensated Heart Failure

**DOI:** 10.3390/diagnostics14182029

**Published:** 2024-09-13

**Authors:** Mihai Grigore, Andreea-Maria Grigore, Adriana-Mihaela Ilieșiu

**Affiliations:** 1Internal Medicine and Cardiology Department, Carol Davila University of Medicine and Pharmacy, “Prof. Dr. Theodor Burghele” Clinical Hospital, 020021 Bucharest, Romania; mihai.grigore@drd.umfcd.ro (M.G.); adriana.iliesiu@umfcd.ro (A.-M.I.); 2Carol Davila University of Medicine and Pharmacy, Cardiology Department Colentina Clinical Hospital, 020021 Bucharest, Romania

**Keywords:** portal vein, congestion, heart failure, mortality, rehospitalization

## Abstract

Background: The severity of systemic congestion is associated with increased portal vein flow pulsatility (PVP). Aim: To determine the usefulness of PVP as a marker of decongestion and prognosis in acute decompensated heart failure (ADHF) patients. Methods: 105 patients, 60% of whom were men, were hospitalized with ADHF, and their PVP index (PVPI) was calculated (maximum velocity–minimum velocity/maximum velocity) × 100 on admission and before discharge, along with their EVEREST score, inferior vena cava diameter (IVC), NT-proBNP, serum sodium, and glomerular filtration rate. A PVPI ≥ 50% was defined as a marker of systemic congestion. After treatment with loop diuretics, a decrease in PVPI of >50% before discharge was considered a marker of decongestion The patients were classified into two groups (G): G1-PVPI decrease ≥ 50% (54 patients) and G2-PVPI decrease < 50% (51 patients). Results: At discharge, compared to G2, G1 patients had lower mean PVPI (14.2 vs. 38.9; *p* < 0.001), higher serum Na (138 vs. 132 mmol/L, *p* = 0.03), and a higher number of patients with a significant (>30%) NT-proBNP decrease (42 vs. 27, *p* = 0.007). PVPI correlated with IVC (r = 0.55, *p* < 0.001), NT-proBNP (r = 0.21, *p* = 0.04), and serum Na (r = −0.202, *p* = 0.04). A total of 55% of patients had worsening renal failure (G1 63% vs. G2 48%, *p* = 0.17). After 90 days, G2 patients had higher mortality (27.45% vs. 3.7 *p* = 0.001) and rehospitalization (49.01% vs. 33.33%, *p* < 0.001). In multivariate regression analysis, PVPI was an independent predictor of rehospitalization (OR 1.05, 95% CI 1.00–1.10, *p* = 0.048). Conclusions: Portal vein flow pulsatility, a meaningful marker of persistent subclinical congestion, is related to short-term prognosis in ADHF patients.

## 1. Introduction

Congestion is the central element in the pathophysiology of heart failure (HF), with over 90% of patient hospitalizations being due to Na and water retention and congestive phenomena [1,2]. It is often clinically underdiagnosed, and its identification remains challenging to quantify because clinical manifestations are the final stage of the pathophysiological processes [3,4,5]. The importance of identifying subclinical congestion is evident as it represents one of the most significant negative predictors influencing the prognosis of HF patients through early rehospitalization and increased mortality [6]. Approximately half of the patients are discharged with subclinical congestion [6,7]. 

Signs and symptoms of congestion are not highly specific for detecting residual congestion. Different methods are used to evaluate venous congestion, with the Venous Excess Ultrasound (VExUS) score being particularly noteworthy. This score assesses venous congestion through Doppler flow patterns of the hepatic, portal, and intrarenal veins, as well as IVC size. In this study, we focus on the portal vein to better understand and assess venous congestion in acute HF patients [8].

Normally, when evaluating PV flow, there is minimal pulsatility often making it very difficult to differentiate between systolic and diastolic components under conditions of normal pressures or minimal variations due to atrial contractility [9,10,11]. In pathological conditions, with the increase in pressure in the right atrium, pressure is transmitted retrogradely into the portal circulation, causing the flow to become pulsatile [9,12,13].

To date, there are limited studies regarding Doppler evaluation of the PV in ADHF patients [11].

The working hypothesis of this study is to assess the usefulness of PVP as a marker for decongestion and short-term prognosis in acute decompensated heart failure (ADHF) patients, as well as for identifying more congested patients upon admission for ADHF.

The specific objectives are to identify the characteristics of ADHF patients with congestion, identify correlation indicators influencing PVPI, mortality, and rehospitalization, but also to assess the impact of PVPI on renal dysfunction and hyponatremia.

## 2. Materials and Methods

### 2.1. Population

This study was conducted as a prospective, observational study at a single center from 1 November 2022 to 1 November 2023.

The diagnosis of ADHF was established according to the 2021 ESC Guidelines for the diagnosis and treatment of acute and chronic HF [14]. All patients included in this study had NT-proBNP levels ≥ 300 pg/dL according to 2021 ESC Guidelines for the diagnosis and treatment of acute and chronic HF [14]. After applying inclusion and exclusion criteria, 105 patients were selected.

The inclusion criteria were as follows:Patients aged > 18 years.Patients with de novo HF, regardless of left ventricular ejection fraction.Patients with recurrent episodes of HF decompensation, regardless of left ventricular ejection fraction.

The exclusion criteria were as follows:Patients in cardiogenic shock.Patients with acute HF associated with acute coronary syndromes, acute myocarditis, and congenital heart diseases.Patients with dyspnea due to concomitant pulmonary diseases: severe pulmonary fibrosis/chronic obstructive pulmonary disease, patients with lung cancer or pneumonia.Patients with severe liver disease (Child-Pugh C).Patients with severe chronic kidney disease (glomerular filtration rate < 15 mL/min/1.73 m^2^) or on dialysis.Patients with severe anemia (hemoglobin < 7 g/dL).Patients with advanced neoplasms and a life expectancy < 1 year.Patients with septic shock.Difficult ultrasound imaging.Pregnant women.

All patients received intravenous loop diuretic treatment within the first 48 h of admission. All patients were examined at two time points: within the first 24 h of admission and before discharge.

PV evaluation was performed at both time moments and, after the second PVPI evaluation, patients were divided into two groups: Group 1 consisted of those who improved PVPI ≥ 50%, and Group 2 consisted of those whose PVPI did not improve <50%. This threshold was selected based on a similar study published in 2020, which demonstrated that patients with poorer prognoses were discharged with more severe pulsatility [15].

All patients provided informed consent, and the study was conducted in accordance with ethical standards, as outlined in the Declaration of Helsinki of 1964 and its subsequent amendments, as well as national recommendations for good medical practice. This study was approved by the Hospital’s ethics committee (9115/26.09.2022).

### 2.2. Clinical and Biochemical Evaluation

After a complete clinical examination, the EVEREST score was calculated (Table 1) [6]. Biochemical tests were collected, including NT-proBNP, hemoglobin, serum Blood Urea Nitrogen (BUN), creatinine, and electrolytes (sodium, potassium), and glomerular filtration rate (GFR) was calculated using the CKD-EPI formula. Worsening renal function (WRF) was defined as an increase in creatinine of more than 0.3 mg/dL between the two time points [1]. A significant decrease in NT-proBNP was defined as a decrease of more than 30% from the initial value [1].

### 2.3. Echocardiography and Portal Vein Ultrasonography

#### 2.3.1. Echocardiography

Echocardiographic examination was performed using a GE Vivid E95 ultrasound machine with a 1.4–4.6 MHz transducer and simultaneous ECG recording. Several parameters were recorded at admission according to the latest EACVI recommendations [16], including left ventricular ejection fraction (LVEF) calculation, left atrial diameter (LAd), and estimation of E/e’ filling pressures. Right heart parameters such as TAPSE, systolic pulmonary artery pressure (PASP), IVC diameter (IVCd)/collapse for PASP estimation, and TAPSE/PASP were also recorded.

#### 2.3.2. Portal Vein Ultrasonography

Portal vein ultrasonography was performed using a Vivid E95 GE Medical System equipped with a 1.5–6 MHz transducer and ECG recording, following recommendations published in the European Heart Journal by Soliman–Aboumarie et al. [9].

To detect the portal vein (PV), patients were positioned in the dorsal decubitus position in post-expiratory apnea. Either the sub-xiphoid or lateral transhepatic windows can be used for imaging. The PV assessment is conducted using color Doppler and Pulsed-Wave Doppler with a baseline velocity of 20 cm/s, synchronized with the electrocardiogram. Maximum (Vmax) and minimum (Vmin) velocities were measured, and the portal vein pulsatility index (PVPI) was calculated using the formula: maximum velocity − minimum velocity/maximum velocity) × 100 (Figure 1). In some cases, tilting the transducer slightly upward from the mid-axillary transhepatic window helped in visualizing the PV more clearly [9].

The quantification of portal vein pulsatility can be achieved through the PVPI. In normal conditions, PVPI <30% and pathologically >50% [11].

Images were analyzed and processed using offline EchoPAC PC v204.

Ultrasonographic evaluation did not influence the treatment decision for patients.

Follow-up of patients was conducted via telephone or by consulting hospital records at 90 days.

### 2.4. Statistical Analysis

The analysis was conducted using IBM SPSS version 20 software for Windows. Descriptive statistics were performed for all patients. Continuous variables were represented as mean with standard deviation if normally distributed, or as median and interquartile range (IQR) if not normally distributed. The normal distribution of data was assessed using the Shapiro–Wilk test. Categorical variables were represented using frequencies (absolute numbers) and percentages.

The correlation between quantitative variables was assessed using Pearson’s or Spearman’s test, while the association between categorical variables was evaluated using the Chi-square test. The Friedman test was used for comparisons within groups across the two time points. Comparisons between groups were conducted using the Student’s *T*–Test for normally distributed data or the Mann–Whitney test for non-normally distributed data.

ROC curve analysis was employed to test the predictive ability of quantitative variables for congestion and decongestion. Multivariate regression analysis was used to identify variables that remained statistically significant predictors of mortality and readmission. Results from multivariate regression were expressed as Odds Ratios (ORs) with a 95% confidence interval.

The significance level for all tests was set at *p* < 0.05.

## 3. Results

A total of 205 patients were hospitalized for ADHF over a period of 12 months. Among them, 105 patients were enrolled (Table 2). Reasons for exclusion included failure to meet the ADHF definition and, most commonly, the inability to obtain adequate imaging (Figure 2).

The median age of the patients was 74 years (IQR 66.5–82 years), with a median left ventricular ejection fraction (LVEF) of 50% (IQR 35–60%). Females accounted for less than half of the patients (40%). There were no statistically significant differences between the two groups regarding age, gender, systolic blood pressure (SBP), heart rate (HR), and LVEF at admission (Table 2).

The main comorbidities included arterial hypertension (HTN), 82.86%, atrial fibrillation (AF), 60%, ischemic heart disease (IHD), 40%, and diabetes mellitus (DM), 39%, with no significant difference between the two groups (Table 2).

Regarding the treatment of HF, the majority of patients were on beta-blocker therapy upon admission, 83%, including 92% in Group 1 and 74% in Group 2 (*p* = 0.01). Mineralocorticoid receptor antagonist (MRA) therapy was present in 70% of patients at admission, including 72% of patients in Group 1 and 68% in Group 2 (*p* = 0.83). As for treatment with ACE inhibitors (ACEis), angiotensin II receptor blockers (ARBs), or angiotensin receptor-neprilysin inhibitors (ARNis), ACEis were the most commonly used, including 32% of all patients, 40% in Group 1, and 23% in Group 2 (*p* = 0.06). Sodium–glucose cotransporter-2 inhibitors (SGLT-2is) were used by 32% of patients overall, including 17% in Group 1 and 15% in Group 2 (*p* = 0.83) (Table 2).

Based on PVPI variations between the two measurements, the patients were classified into two groups: Group 1-PVPI variation ≥50% 54 patients, median age 72.5 years and Group 2-PVPI variation < 50% 51 patients, median age 75 years (Table 3).

Upon admission, the mean EVEREST score was 12 (±4), median NT-proBNP was 4060 pg/mL (1865–7085), IVCd was 22 (19–24) mm, and the PVPI was 50.6% (±22.9), without significant difference between Group 1 and Group 2 for any of the parameters (Table 3).

At discharge, the EVEREST score decreased from 11 (±4) to 2 (±2) without significant difference between groups.

The mean PVPI was significantly lower in Group 1 compared to Group 2 (14.2 vs. 38.9; *p* < 0.001) (Table 3).

### 3.1. PVPI and Commonly Used Congestion Parameters

We correlated the PVPI value at admission with other clinical, biological, and echocardiographic parameters commonly used for congestion analysis. The PVPI showed a positive correlation with the EVEREST score (r = 0.366, *p* < 0.001), NT-proBNP (r = 0.275, *p* = 0.005), and IVCd (r = 0.355, *p* = 0.002), and a negative correlation with the TAPSE/PASP ratio (r = −0.324, *p* = 0.001).

The median E/e’ ratio at admission was 28.53, with no significant difference between the two groups: 27.45 in Group 1 and 28.9 in Group 2 (*p* = 0.41). Upon discharge, there was a decrease in the E/e’ ratio to 14.82, though this reduction was not statistically significant. In Group 1, the E/e’ ratio decreased to 13.71, and, in Group 2, it decreased to 16 (*p* = 0.06) (Table 3).

The PVPI value at discharge correlated with clinical, biochemical, and echocardiographic parameters, documented at discharge, which are commonly used in predicting decongestion: EVEREST score (r = 0.389, *p* < 0.001), NT-proBNP (r = 0.210, *p* = 0.047), and IVCd (r = 0.554, *p* < 0.001), and negatively with TAPSE/PASP (r = −0.373, *p* = 0.014). ROC curve analysis tested several parameters in predicting PVPI improvement. Among these, IVC diameter and PASP at discharge predicted PVPI improvement with very good accuracy (AUC = 0.710, *p* = 0.002; AUC 0.743, *p* < 0.001).

At admission, the mean PVPI was 50.98% (21.99) and decreased at discharge to 25.2% (20.24) *p* < 0.001 (Figure 3). All patients had abnormal PVPI at inclusion.

Regarding NT-proBNP, in Group 1, there were more patients with a > 30% decrease in NT-proBNP at discharge (42 vs. 12 patients, compared to 27 vs. 24), *p* = 0.001 (Table 4). The absolute value of NT-proBNP was not significantly different between the two groups at discharge (*p* = 0.445).

### 3.2. PVPI in Patients with Renal Dysfunction and Hyponatremia

The median glomerular filtration rate was 54 mL/min/1.73 m^2^ (40.5–71) at admission and was similar between the two groups. WRF occurred in 58 patients (55.23%), with an incidence of 62.74% in Group 1 compared to 48.14% in the other group (*p* = 0.17) (Table 4).

The mean Na level at admission was 138.86 mmol/L (3.73) and was similar between the two groups (*p* = 0.14). The Na level at admission was statistically significantly correlated with the PVPI value, indicating that hyponatremia was more frequent in patients with increased PVPI (Group 2) (r = −0.202, *p* = 0.04). At discharge, patients from Group 2 had significantly lower Na values: 132.58 vs. 138 mmol/L (*p* = 0.03).

### 3.3. Hospital Rehospitalization and Mortality

During hospitalization and within the first 90 days after discharge, 16 patients (15.23%) died and 43 patients (40.95%) were readmitted due to ADHF. Death and rehospitalization rates were significantly higher in patients from Group 2, who did not achieve a ≥50% improvement in PVPI (rehospitalization: 49.01% vs.33.33%, *p* = 0.043; death: 27.45% vs. 3.7%, *p* = 0.001).

Several parameters were predictors for rehospitalization: IVCd (AUC = 0.709, *p* = 0.002, cut-off 14.5, sensitivity 86.5%, specificity 77.5%), PASP (AUC = 0.761, *p* = 0.001, cut-off 25.55, sensitivity 89.2%, specificity 70%), NT-proBNP (AUC = 0.732, *p* < 0.001, cut-off 519.5, sensitivity 82%, specificity 71%), and PVPI at discharge (AUC = 0.752, *p* < 0.001, cut-off 34.4, sensitivity 94.6%, specificity 72.5%) (Figure 4).

There was a statistically significant association between >50% improvement in PVPI at discharge and patient death at 3 months (Chi-square = 11.45. *p* = 0.001) and in-hospital mortality (Chi-square = 7.94, *p* = 0.005).

Among the parameters evaluated for decongestion, after introducing them into multivariate regression, PVPI was the only parameter that remained statistically significant (OR = 1.05, 95% CI 1.01–1.10, *p* = 0.048).

## 4. Discussion

Our study aimed to investigate the patterns of PVP alongside the decongestion process in ADHF patients. Among our patients admitted with ADHF, all exhibited abnormal PVP profiles, which notably improved following diuretic treatment guided by clinical assessment. Remarkably, we found that PVPI improvement was associated with favorable outcomes including, decreased hospital rehospitalization within a three-month period.

Prior studies have demonstrated that higher portal vein pulsatility at discharge is associated with an unfavorable prognosis in patients with HF [13,17]. This is consistent with our study, where rehospitalization rates were 49.01% versus 33.33% (*p* = 0.043) and death rates were 27.45% versus 3.7% (*p* = 0.001) in patients without PVPI improvement.

Furthermore, a recent study demonstrated that, in patients with ADHF and congestion following an acute event, the presence of moderate/severe PVPI at discharge was linked to elevated all-cause mortality [11]. This finding aligns with our study, which showed death rates of 27.45% versus 3.7% (*p* = 0.001) in patients without PVPI improvement.

The clinical examination of HF patients is essential; however, signs and symptoms typically manifest only when the congestion is moderate to severe. Although these clinical parameters are well-described and familiar to physicians, they have low sensitivity in detecting less severe congestion. In our study, the EVEREST score, used to clinically quantify signs and symptoms of congestion, decreased in both groups, but the difference between the two groups was not statistically significant. The literature has primarily focused on the prognostic value of clinical scores rather than their diagnostic utility, revealing an associated 15% increased risk of mortality in patients with HF [6].

We observed statistically significant correlations between PVPI and the most known parameters: EVEREST score, NT-proBNP, and IVDd.

In this study, although the average dose of oral diuretics before hospital admission was similar between the two groups, patients in Group 2 required higher doses of intravenous loop diuretics during hospitalization. The diuretic doses were determined by the attending physician and were comparatively lower than those reported in other studies, such as Meani et al. (2023) [18]. Our findings are consistent with the existing literature, which shows an association between the need for higher diuretic doses in HF patients with severe congestion and an increased incidence of adverse events and poorer prognosis [18]. This association likely reflects more advanced HF rather than a direct harmful effect of aggressive diuretic therapy, as patients in Group 2 had significantly higher NT-proBNP levels and more frequent hyponatremia, and there was no significant negative impact on renal function.

The majority of patients were on beta-blockers (83%), although there was a significantly lower number of patients in Group 2 (74%) compared to Group 1 (91%). This likely indicates that the more severe congestion was associated with either intolerance or contraindications to beta-blocker therapy.

Concerning RAAS inhibition, more patients were treated with ACE inhibitors in Group 1 and a higher number of patients were on ARBs in Group 2. Therefore, the overall proportion of patients receiving RAAS inhibitors in the two groups was comparable (52% in Group 1 compared to 44% in Group 2).

NT-proBNP has proven to be an effective biomarker for assessing decongestion, in accordance with the literature data demonstrating that a reduction of more than 30% before discharge is associated with effective decongestion [1]. In our study, the decrease in PVPI before discharge was correlated with a more than 30% reduction in NT-proBNP levels. Of note, the mean NT-proBNP values did not significantly differ between the two groups. In previous studies showing the value of PVPI in assessing the risk of mortality and rehospitalization in HF patients, the reported NT-proBNP value was 623 pg/dL [13]. In our study, the patient cohort had an average NT-proBNP value of 4060 pg/dL, indicating more severe congestion associated with HF.

In the management of congestion there is a delicate balance between achieving clinical improvement and the occurrence of adverse effects associated with diuretic therapy, particularly WRF and electrolyte imbalances [19].

In our study, there were differences in WRF based on the variation in PVPI. WRF occurred in 58 patients (55.23%) and was non-significantly more frequent in Group 1, 62.74%, compared to 48.14% in Group 2 (*p* = 0.17). These results are in agreement with the literature data indicating that WRF diagnosed solely through consecutive serum creatinine measurements does not independently determine outcomes in ADHF patients [19]. If WRF is transient and is associated with responsive congestion to diuretics (so-called „pseudoWRF”), it does not affect the patient’s prognosis. On the contrary, persistent WRF associated with persistent congestion („true WRF”) worsens patients’ prognosis [20].

It has been proposed that PVPI is useful as a diagnostic tool and as a marker of therapeutic efficacy for hypervolemic hypotonic hyponatremia [21]. In our study, patients who did not demonstrate improvement in PVPI were more likely to experience hyponatremia at discharge. Patients from Group 2 had significantly lower Na values: 132.58 mmol/L versus 138 mmol/L (*p* = 0.03).

The present study aimed to investigate the patterns of PVP alongside the decongestion process in ADHF patients. Our results demonstrated that PVPI is a highly valuable marker for assessing the severity of congestion and for the evaluation of decongestion upon hospital discharge.

To date, there is no single parameter for the assessment of systemic decongestion in HF, emphasizing the need for a multiparametric approach. In the assessment of systemic venous congestion by quantifying the flow pulsatility in the portal vein, along with the inferior vena cava, hepatic, and intra-renal vein, Doppler ultrasound should be complementary to the clinical signs of congestion, which have low specificity and sensitivity. An advantage of PVP evaluation is the significantly faster learning curve compared with, for example, the evaluation of intrarenal venous flow.

Until now, most studies have analyzed the pulsatility of the portal vein flow as a congestion parameter in postoperative patients. The novelty of our study is proving the diagnostic and prognostic role of portal vein flow pulsatility as a congestion marker in patients with acute decompensation of chronic HF.

## 5. Limitations

This is a non-randomized study conducted at a single center with a relatively small patient population and a brief follow-up period. Further studies with a larger number of patients are needed.

## 6. Conclusions

PVPI is a valuable parameter for the assessment of congestion in HF and is a useful discriminator of persistent subclinical congestion. As an independent factor associated with better clinical outcomes, it should be incorporated into the evaluation algorithm of systemic decongestion.

## Figures and Tables

**Figure 1 diagnostics-14-02029-f001:**
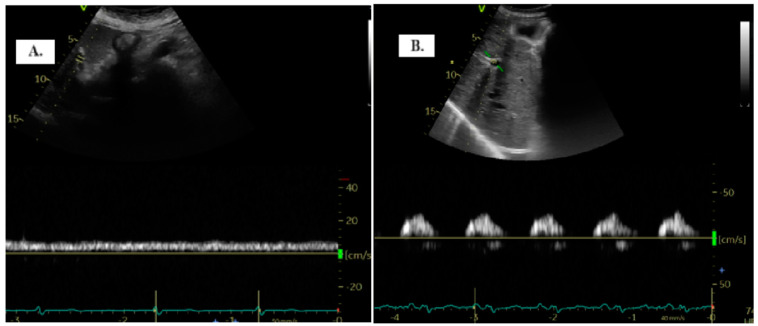
The two distinct types of portal vein Doppler patterns: continuous (**A**) and discontinuous (**B**).

**Figure 2 diagnostics-14-02029-f002:**
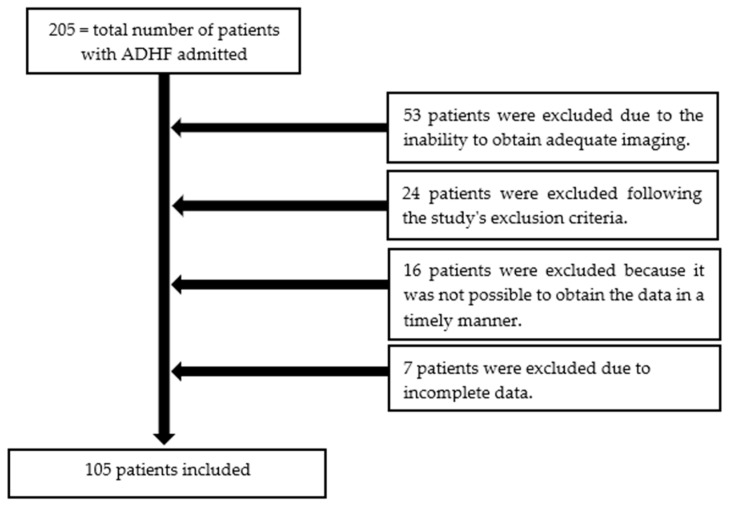
The patient selection process from this study.

**Figure 3 diagnostics-14-02029-f003:**
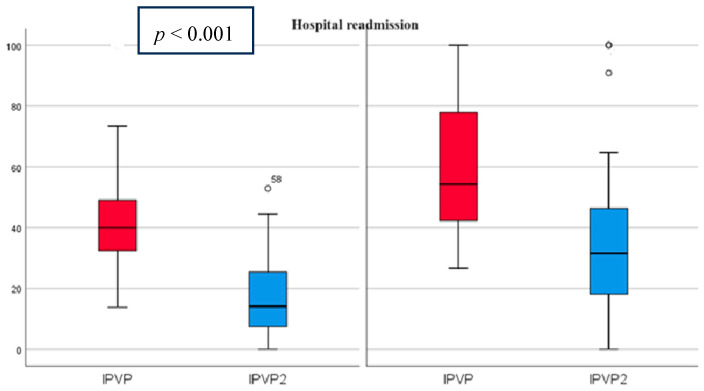
PVPI on admission and discharge in predicting patient readmissions PVPI (red)—admission, PVPI2 (blue)—discharge. On the left side are patients without rehospitalization and on the right side are patients with rehospitalization, showing that PVPI is higher both on admission and at discharge in patients with rehospitalization.

**Figure 4 diagnostics-14-02029-f004:**
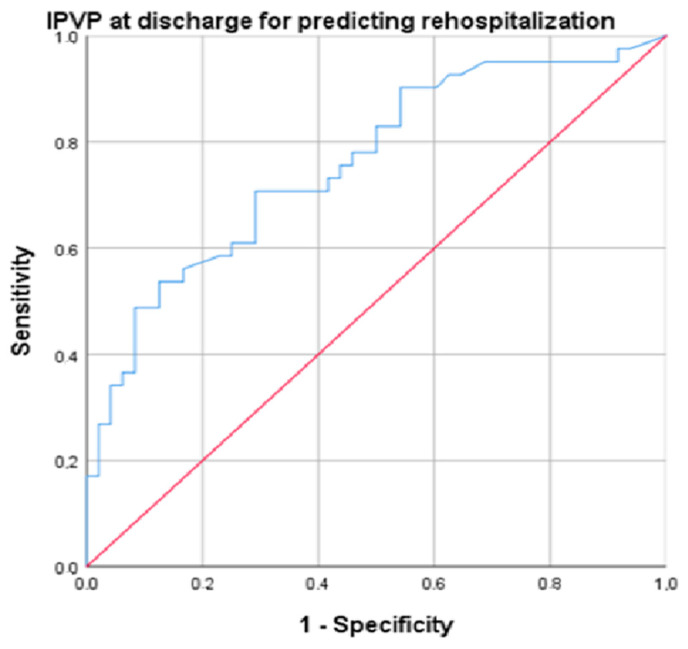
ROC (Receiver Operating Characteristic) curve for the analysis of PVPI at discharge for predicting rehospitalization (blue line). Diagonal (red) line represents a random classifier, where the TPR (true positive rate) equals the FPR (false positive rate).

**Table 1 diagnostics-14-02029-t001:** The EVEREST score [6].

Signs/Symptoms	Scale
Dyspnea	0 = None; 1 = Seldom; 2 = Frequent; 3 = Continuous
Ortopnea	0 = None; 1 = Seldom; 2 = Frequent; 3 = Continuous
Jugular venous distention	0 = ≤6; 1 = 6–9; 2 = 10–15; 3 = ≥ 15
Rales	0 = None; 1 = Bases; 2 = Up to < 50%; 3 ≥ 50%
Edema	0 = Absent/trace; 1 = Slight; 2 = Moderate; 3 = Marked
Fatigue	0 = None; 1 = Seldom; 2 = Frequent; 3 = Continuous

**Table 2 diagnostics-14-02029-t002:** Clinical characteristics of patients according to PVPI changes.

Baseline	Total (105 Patients)	Group 1 (54 Patients)	Group 2 (51 Patients)	*p* Value
Age, years, median (IQR)	74 (66.5–82)	72.5 (63–80)	75 (68–84)	0.17
Sex, male (%)	63 (60%)	30 (56%)	33 (65%)	0.42
HR, beats/min	76 (70–100)	77.5 (70–110)	75 (65–100)	0.18
SBP, mmHg, median (IQR)	130 (115–147)	130 (115–150)	130 (110–146)	0.71
IHD, (%)	40 (38.1%)	16 (29.63%)	24 (47.06%)	0.74
HTN, (%)	87 (82.86%)	44 (81.48%)	43 (84.31%)	0.7
DM (%)	39 (37.14%)	21 (38.89%)	18 (35.29%)	0.7
AF (%)	60 (57.14%)	31 (57.41%)	29 (48.33%)	0.9
LVEF (%)	50 (35–60)	50 (36–65)	50 (34–60)	0.95
ACEi admission (%)	34 (32%)	22 (40%)	12 (23%)	0.06
ARB admission (%)	18 (17%)	7 (12%)	11 (21%)	0.3
ARNI admission (%)	18 (17%)	9 (16%)	9 (17%)	1
Beta-blocker admission (%)	88 (83%)	50 (92%)	38 (74%)	0.01
MRA admission (%)	74 (70%)	39 (72%)	35 (68%)	0.83
SGLT2i admission (%)	34 (32%)	18 (17%)	16 (15%)	0.83
Furosemide before admission (mg/day), median (IQR)	20 (0–40)	20 (0–35)	20 (0–40)	0.06
Furosemide in hospital (mg/day), median (IQR)	60 (40–80)	60 (40–80)	80 (40–80)	0.02

HR—heart rate; SBP—systolic blood pressure; IHD—ischemic heart disease; HTN—arterial hypertension; DM—diabetes mellitus; AF—atrial fibrillation, LVEF—left ventricular ejection fraction; ACEi—Angiotensin-converting-enzyme inhibitors. ARB—Angiotensin II receptor blockers. ARNI—angiotensin receptor receptor-neprilysin inhibitor. MRA—mineralocorticoid receptor antagonists, SGLT2—sodium-glucose cotransporter-2.

**Table 3 diagnostics-14-02029-t003:** Clinical, biochemical, and echocardiographic parameter variation from admission to discharge.

Parameters	Total (105 Patients)	Group 1 (54 Patients)	Group 2 (51 Patients)	*p* Value	Total (105 Patients)	Group 1 (54 Patients)	Group 2 (51 Patients)	*p* Value
**Time**	**Admission**				**Discharge**			
EVEREST score, median	12 (8–14)	12 (8–14)	11 (9–15)	0.23	2 (2–3)	2 (2–3)	2 (2–4)	0.50
Weight (kg), mean	84.47 (±18.16)	83.63 (±17.5)	84.25 (±19.29)	0.87	82.41 (±17.33)	81.83 (±16.86)	83.06 (18.01)	0.72
GFR, (mL/min)	54 (40.5–71)	55 (45–71)	52.5 (34–69)	0.08	58.53(43.72–74.61)	58.13 (43.14–67.16)	61.84 (44.25–80.48)	0.43
NT-proBNP (pg/mL)	4060 (1865–7085)	4502 (2461–7085)	3283 (1198–10839)	0.27	1537 (926–2940)	1476 (913–2741)	1672 (1013–3912)	0.44
Na (mmol/L)	138.86 (±3.73)	139.46 (±3.82)	138.28 (±3.56)	0.14	135.76 (±15.14)	138 (±3.33)	132.58 (±23.08)	0.03
E/e’ median	28.53 (20–37.12)	27.45 (18.62–37.12)	28.9 (23.87–38.66)	0.41	14.82 (11.31–23.87)	13.71 (10.22–18.25)	20 (13.55–27)	0.06
TAPSE (mm)	17 (14–19)	17 (14–19)	17 (14–18)	0.78	17 (14–19)	17 (14–20)	16 (15–18)	0.24
IVC (mm)	22 (19–24)	21(19–23)	22(18.5–25)	0.08	17 (15–22)	16 (15–19)	20.5 (16–23)	0.002
PASP (mm Hg)	50.89 (±14.57)	49.73 (±14.63)	52.18 (±14.54)	0.39	38.16 (±12.49)	33.43 (±9.72)	45.11 (±12.98)	< 0.001
TAPSE/PASP	0.35	0.36	0.34	0.61	0.43	0.37	0.29	0.08

GFR—glomerular filtration rate; Na—Natrium; TAPSE—tricuspid annular plane systolic excursion; IVC—inferior vena cava; PASP—pulmonary arterial systolic pressure.

**Table 4 diagnostics-14-02029-t004:** Variation in biochemical markers, rehospitalization, and mortality.

Outcome	Total (105 Patients)	Group 1 (54 Patients)	Group 2 (51 Patients)	*p* Value
WRF	58 (55.23%)	26 (48.14%)	32 (62.74%)	0.17
NT-proBNP drop > 30%	69 (65.7%)	42 (77%)	27 (52.9%)	0.01
Rehospitalization %	43 (40.95%)	18 (33.33%)	25 (49.01%)	0.043
Death, number, %	16 (15.23%)	2 (3.7%)	14 (27.45%)	0.0018

WRF—worsening renal function.

## Data Availability

Data used in this study may be provided by the authors upon reasonable request.

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
