# Peer review of "Portal Vein Pulsatility: A Valuable Approach for Monitoring Venous Congestion and Prognostic Evaluation in Acute Decompensated Heart Failure"

_diagnostics, 2024, doi:10.3390/diagnostics14182029_

Round 1

Reviewer 1 Report

Comments and Suggestions for Authors

In the present study, Grigore M et al. conduct a prospective observational study on a cohort of 105 patients admitted for acute heart failure, in which they analyze the potential utility of portal Doppler variation to identify patients with worse prognosis and its relationship with other clinical and analytical markers of congestion.

The study has a sufficient sample size, and the topic proposed by the authors is highly relevant in the field of heart failure. However, before the paper can be accepted, I believe it is necessary to clarify/resolve a few issues.

1.Introduction

The authors begin the work by discussing the importance of residual congestion and its high prevalence at discharge in acute heart failure. However, they then shift the focus to ultrasound measures of intravascular congestion, which somewhat disrupts the continuity of the argument. In the current context, a study on the assessment of congestion in heart failure patients should be more specific about the type of congestion being measured. Additionally, since portal vein analysis is part of the VExUS protocol, I believe it should be mentioned in this section.

I would suggest that the authors be more explicit about the hypothesis they aim to demonstrate. For example, "Our working hypothesis is that PVPI allows for the identification of more congested patients upon admission for acute heart failure."

2.Material and methods

i. In the inclusion criteria, was the NT-proBNP cutoff based on the guidelines? This should be made clear.
ii. Clearly define what constitutes severe chronic kidney disease, severe liver disease, and severe anemia (cutoff points).

3. Results

If 205 patients were included in the initial analysis and ultimately 105 were included, a chart or figure should be provided showing the reasons and the approximate percentage for each type of exclusion.

Table 2: Since this is a study on congestion, it is crucial to know the prior oral diuretic dose. Patients in group 2 (absence of improvement in PVPI) might have been on higher doses of outpatient diuretics and could present with diuretic resistance upon admission, which would introduce selection bias. The same applies to in-hospital treatment. Although diuretic doses were decided by physicians according to guidelines, it would be important to know the doses the patients received, especially considering that admission and discharge are being compared. This could represent a significant source of bias.

Regarding kidney function, the observed results support the current thesis about pseudo-WRF, patients in group 1 had a worsening of kidney function, and despite everything they became more decongested.

Reviewer 2 Report

Comments and Suggestions for Authors

The manuscript addresses an important topic in cardiology. However, many things need to be improved to make it suitable for publication.

The spelling should be checked to be more in line with the medical-scientific style (e.g. biological analysis).

The choice of a PVPI cut-off of 50% to divide the groups should be explained.

Clarify the inclusion/exclusion criteria. What is severe anemia or severe sepsis? 

The abstract needs to be rewritten with a clearer presentation of the study groups.

What was the outcome parameter? If PVPI more or less 50% at discharge, this should be moved to the method section. And after forming two groups (after discharge), your study has started. So the aim was to evaluate the short-term prognostic power of PVPI in this cohort of patients.

If you are explaining subclinical congestion/monitoring decongestion at discharge based on PVPI change compared to admission, please include other signs of CHF compensation, such as weight loss, regression of peripheral edema. 

In the Discussion section, please provide more details on how your study compares to other similar studies (e.g., 10.4330/wjc.v15.i11.599).

Please discuss all your results, including differences in treatment with beta-blockers and ACE inhibitors in two groups.

Please, highlight the novelty of your study comparing many others.

Comments on the Quality of English Language

English should be significantly improved.

Round 2

Reviewer 1 Report

Comments and Suggestions for Authors

The authors have made the suggested changes and the manuscript has been substantially improved.

 I would suggest that the authors refer in the discussion section to the poor prognosis of high-dose diuretics. According to their results, patients with greater congestion detected by portal Doppler required higher doses of intravenous diuretics during admission, and this group had a higher rate of events.

Author Response

Comment 1: I would suggest that the authors refer in the discussion section to the poor prognosis of high-dose diuretics. According to their results, patients with greater congestion detected by portal Doppler required higher doses of intravenous diuretics during admission, and this group had a higher rate of events.

Response 1:

Thank you for your valuable feedback and for highlighting the need to address the poor prognosis associated with high-dose diuretics. We agree that this aspect required further elaboration, and we have revised the discussion accordingly to better clarify this point. In our revised manuscript, we have emphasized that patients with greater congestion, as detected by portal Doppler, required higher doses of intravenous diuretics during hospitalization, which is associated with an increased rate of adverse events and poorer prognosis.

The revised text now reads:

"In this study, although the average dose of oral diuretics before hospital admission was similar between the two groups, patients in Group 2 required higher doses of intravenous loop diuretics during hospitalization. The diuretic doses were determined by the attending physician and were comparatively lower than those reported in other studies, such as Meani et al. (2023) [19]. Our findings are consistent with existing literature that shows an association between the need for higher diuretic doses in HF patients with severe congestion and an increased incidence of adverse events and poorer prognosis [19]. This association likely reflects more advanced HF rather than a direct harmful effect of aggressive diuretic therapy, as patients in Group 2 had significantly higher NT-proBNP levels, more frequent hyponatremia, and there was no significant negative impact on renal function."

These changes have been made in the Discussion section on [page number 9], [paragraph number 6], [line number 295-305] of the revised manuscript.

We have also included an additional reference to support this point:

Meani P, Pagnoni M, Mondellini GM, et al. Impact of loop diuretic dosage in a population of patients with acute heart failure: a retrospective analysis. Front. Cardiovasc. Med. 10:1267042, 2023. doi: 10.3389/fcvm.2023.1267042.

Thank you again for your insightful comments, which have helped improve the clarity and depth of our discussion.

Reviewer 2 Report

Comments and Suggestions for Authors

All corrections have been done in accordance to recommendations. Congrats!

Author Response

Thank you for your positive feedback and for acknowledging the corrections made to the manuscript. We appreciate your valuable suggestions and are pleased that the revisions align with your recommendations.